

# Vitamin D supplementation in Saudi Arabia: mothers' knowledge, practices, and attitudes

Hanan F. Alharbi[1] and Khulud Ahmad Rezq[2]

[1] Maternity and Pediatric Nursing Department, College of Nursing, Princess Nourah bint Abdulrahman University, Riyadh, Saudi Arabia
[2] Community and Psychiatric Health Nursing Department, Faculty of Nursing, University of Tabuk, Tabuk, Saudi Arabia

Corresponding author
Hanan F. Alharbi,
hfalharbi@pnu.edu.sa

## ABSTRACT

**Objective**. Despite abundant sunshine, vitamin D (VIT D) deficiency remains prevalent in tropical regions such as the Kingdom of Saudi Arabia (KSA), largely due to limited dietary intake and sun exposure restrictions. This study evaluated mothers' perceptions of VIT D supplementation for them and their children in KSA.

**Methods**. This observational cross-sectional study was conducted in KSA among mothers aged 18 and older with children under five years old. A convenience sampling method was employed to recruit participants in the study in the period from 31 July to 2 October 2022. The data were collected by developed questionnaire which was distributed to participants via WhatsApp and Facebook. The data were analyzed using an SPSS program with both descriptive and inferential statistical methods.

**Results**. A total of 200 mothers participated, with a mean age (SD) of 35.48 (6.9) years. Among them, 66.40% had poor knowledge of VIT D and its supplementation, while 33.60% demonstrated good knowledge. Regarding practices, only 22.95% of mothers reported good practices, 43.85% reported fair practices, and 33.20% reported poor practices. However, most mothers (95%) demonstrated a favorable attitude toward VIT D and its supplementation, and only 5% of them showed a negative attitude. Notably, significant correlations were observed between mothers' salaries and their familiarity with VIT D ($p = 0.024$); their age and practices towards VIT D and its supplements ($p = 0.049$); as well as their level of education, functional status, and attitudes towards VIT D ($p = 0.037$ and $p = 0.008$, respectively).

**Conclusion**. The findings indicate that a significant proportion of mothers had poor knowledge and inadequate practices regarding VIT D and its supplementation, despite their generally positive attitudes. These results underscore the need for targeted interventions to enhance knowledge and practices, considering socio-economic factors and educational background.

## INTRODUCTION

Vitamin D (VIT D), a fat-soluble vitamin, plays a key role in regulating the absorption and retention of essential minerals such as calcium and phosphorus, which are crucial for bone development (*Wimalawansa, 2024*). Sunlight exposure serves as the primary source
of VIT D, while certain foods, including egg yolks, tuna, sardines, salmon, and cod liver oil, are also rich in this vitamin (*Bischofova et al., 2018*). Notably, human milk contains lower levels of VIT D compared to animal milk (*Elnagar, Abd El Salam & Abdel-Sadik, 2020*).

A deficiency in VIT D among newborns and adolescents can lead to conditions such as hypocalcemic seizures or convulsions. Additionally, infants and young children with VIT D deficiency may develop bone deformities or rickets (*Bassam & Abd-Elmageed, 2022*).

Moreover, VIT D deficiency has been associated with the development of cardiomyopathy, which can lead to heart failure and even death (*Elnagar, Abd El Salam & Abdel-Sadik, 2020*; *Vincent, 2025*).

VIT D deficiency can commonly occur among children with ages ranging 5 to 10 years due to various factors such as inadequate VIT D-rich food intake, limited sunlight exposure, wearing clothing that covers the entire body, ethnicity, and the VIT D status of their mothers during pregnancy (*Ahmed et al., 2017*). This deficiency is a global issue, and VIT D insufficiency or deficiency affects at least 75% of the global population (*Siddiqee et al., 2021*). It has been reported that approximately 254 million children worldwide suffer from VIT D deficiency (*Abbas, Kassim & Hameed, 2022*). Surprisingly, even in countries with ample sunlight, such as the Kingdom of Saudi Arabia (KSA) and Oman, the prevalence of VIT D deficiency remains high (*Kavitha, 2015*). Reduced ultraviolet (UV) light exposure amplifies the severity of VIT D deficiency.

A study conducted in Saudi Arabia on the prevalence of VIT D deficiency in children under two years of age indicated that the condition is common among this age group, particularly in infancy (*Al-Qahtani et al., 2022*). A recent systematic review in Saudi Arabia showed a high prevalence of VIT D deficiency among children and adolescents (*Alahmadi, Morsi & AlNagshabandi, 2020*).

VIT D deficiency remains prevalent in tropical countries such as Saudi Arabia, primarily due to insufficient intake of VIT D-fortified or VIT D-rich foods, as well as social customs that limit sunlight exposure, despite the abundance of sunlight (*Pettifor, 2014*). Another study suggests this deficiency contributes to several health conditions such as osteoporosis, anemia, and inflammatory bowel (*Kumar et al., 2024*).

Saudi Arabian national guidelines have recommended supplementation and food fortification programs for VIT D deficiency. The Saudi Food and Drug Authority (SFDA) recommends VIT D and other minerals be added to specific food products for high-risk groups such as infants, children, and older adults (*SFDA, 2022*). However, despite these efforts and recommendations, VIT D deficiency remains prevalent, with a recent review indicating that about 80% of the population has inadequate levels of VIT D (*Alahmadi, Morsi & AlNagshabandi, 2020*).

Lack of awareness is recognized as a significant contributing factor to this deficiency, emphasizing the importance of community-level awareness and educational campaigns to prevent serious adverse events associated with VIT D deficiency (*Ahmed, 2015*; *Anjorin, 2020*). Maternal education is strongly linked to children's health, and mothers' nutritional awareness also plays a crucial role in promoting children's well-being. A mother's nutritional and healthcare literacy significantly impacts the level of care she provides (*Kaur, Grover & Kaur, 2015*). Therefore, health education initiatives should prioritize

![PeerJ]

equipping mothers with knowledge about VIT D, including its importance for health, the risks of deficiency, and preventative measures (*Abate, Murugan & Gualu, 2016*; *Scotti et al., 2019*).

To address the need for improved knowledge and awareness of the importance of VIT D and its deficiency, it is essential to explore mothers' perceptions of VIT D supplementation in countries like Saudi Arabia. Unfortunately, there are no studies investigating mothers' perceptions of VIT D in this context. Thus, the current study was designed to fill this knowledge gap by examining mothers' perceptions of VIT D supplementation for them and their children in Saudi Arabia.

## METHODS

### Study design and sample

This is a cross-sectional study carried out in Saudi Arabia and involving a sample of mothers who were older than 18 years old with children under the age of 5 years. Only individuals who consented were included. Mothers younger than 18 years old and those with youngest children older than 5 years were excluded. The selection of mothers of children aged under 5 years for this study was decided to ensure that high-risk mothers were included and to cover the gap in knowledge since most previous studies were conducted on mothers of infants or children between 6–17 years (*Al-Othman et al., 2012*).

### Instrument

This study utilized an online questionnaire that was adapted from a previous study (*Kavitha, 2015*) with approval obtained from the author of the tool. To ensure content validity, a panel of nursing experts was consulted, including an associate professor in Medical-Surgical Nursing Department, an associate professor in Maternity Health Nursing Department, and an associate professor in Community Nursing Department from two different governmental universities.

To assess clarity and the participant experience, the researchers piloted the survey with a sample of 20 participants. The final questionnaire, initially developed in English and later translated into Arabic, was estimated to take 10–15 min to complete. The online questionnaire consisted of five sections designed to evaluate the knowledge, attitudes, and practices of mothers regarding VIT D. Part I focused on the sociodemographic characteristics of the participating mothers, such as educational level, age, and number of children. Part II examined mothers' knowledge of VIT D and consisted of six questions that assessed their understanding of the sources and benefits of VIT D. Part III explored mothers' practices related to VIT D and its supplementation for their children. It included nine questions concerning their habits of exposing their children to sunlight and their usage of VIT D supplements. Part IV examined mothers' attitudes toward VIT D and its supplementation. This section was comprised of seven questions regarding their opinions on using VIT D supplements during pregnancy and breastfeeding, as well as their views on regularly checking VIT D levels in their bodies. Part V examined mothers' practices concerning the use of VIT D, encompassing 13 questions related to their consumption of VIT D and other related practices.
## Data collection

A Google Form was employed to create a data collection tool, which was subsequently disseminated *via* social media channels such as WhatsApp and Facebook. The link was accompanied by a concise explanation encouraging mothers to participate. Multiple reminders were sent throughout the data collection period, which spanned from 31 July to 2 October 2022. The link was shared with mothers by sending a link of the tool by their colleagues and friends. The link was anonymous and shared with the mothers asking them to respond to the survey. The link was available for mothers to respond once using their email address. If two or more responses were reported, they were deleted from the data collection sheet.

## Scoring system

The knowledge section employed a binary scoring system, awarding one point for each correct answer and zero points for incorrect responses. If the total score exceeded 50%, this indicated a good level of knowledge. Conversely, a total score below 50% indicated poor knowledge. The potential range of correct answers in this section ranged from 0 -18. For the practice questions, scores were assigned based on the adoption of healthy practices by the mothers to enhance their VIT D levels. Each correctly answered question was scored as 1, while an incorrect answer was awarded 0. The range of correct answers in this section spanned from (0–8). To assess adherence to good practices, the study established a cut-off score of 75% for the practice section. Scores exceeding this threshold were considered indicative of good practice. Scores falling below 50% were considered indicative of poor practice. Scores between these two thresholds reflected fair practice (between 50–75%). Regarding the attitude questions, a score of 1 was assigned for a "yes" response, while a score of 0 was given for a "no" response. The total attitude score ranged from (0–6). A positive attitude was defined as a total score exceeding 75%, while a score between 50% and 75% indicated a neutral attitude. A score below 50% indicated a negative attitude.

## Sample size

A statistical power analysis determined the minimum required sample size using G*Power 3.1 software (*Faul et al., 2007*).

The power calculation in this study was based on the null hypothesis ($H_0$) that there is no significant difference between mothers' VIT D knowledge, attitudes, and practices and demographic variables, and the alternative hypothesis ($H_1$) that a significant effect exists between dependent and independent variables. The study included a total of 200 mothers who responded to the distributed link. However, to account for potential participants not meeting the inclusion criteria, the sample size was increased by 20%. Therefore, the statistical power analysis yielded a minimum sample size of 191 participants.

## Data analysis

This analysis was conducted using SPSS version 22 (IBM Corp., Armonk, NY, USA). Descriptive statistics, including percentages and frequencies, were employed to analyze the categorical data. Kolmogorov–Smirnov and Shapiro–Wilk tests were used to assess the normality of data distribution for the continuous variables. As variables were not

normally distributed, data were analyzed using the median and interquartile range (IQR). A logistic regression model was used to identify the relationship between participant characteristics and mothers' knowledge, practices, and attitudes. A $P$-value $\leq 0.05$ was considered significant.

### Ethical approval

The research protocol underwent ethical review and approval by the relevant Institutional Review Board (IRB) at Princess Nourah bint Abdulrahman University in Riyadh, Saudi Arabia (HAP-01-R-059). Before participating in the survey, mothers were provided with information regarding their rights to voluntarily participate or withdraw from the study. Informed consent was obtained at the beginning of the survey, where participants were asked to indicate their consent by clicking either "yes" or "no" before proceeding to answer the survey questions. To ensure the confidentiality of participant information, all data were stored securely on the primary researcher's computer, and only the researchers involved in this study had access to the data.

## RESULTS

### General and demographic characteristics of the included participants

A total of 200 mothers with a median age of 33 (IQR = 8) were included in the study. Most of the mothers were married (97.5%), had a bachelor's degree (63%), were unemployed (63%), Saudi citizens (75.6%), home-owning (57%), and had monthly incomes ranging from 5 to less than 10,000 riyals per month (40.5%). Regarding their offspring, a significant proportion of mothers (35.5%) had four or more children, with most mothers (40.5%) having youngest children aged one to 12 months. Bottle feeding was the predominant method of feeding among mothers (52.5%) (Table 1).

### Mothers' knowledge about VIT D

Nearly all mothers (99%) reported awareness of VIT D. Healthcare professionals (66.0%) and social media (43.5%) were the primary sources of information about VIT D. Regarding dietary sources, mothers most correctly identified milk/dairy, fish, egg yolks, and liver as food sources of VIT D. Regarding other sources of VIT D, most respondents (83%) reported the sun as a primary source. However, a relatively lower percentage acknowledged the role of food (20%) and food supplements (25.0%) as additional sources of VIT D. In addition, the respondents demonstrated good awareness about the benefits of VIT D. The majority recognized its positive impact on bone health (88%), immunity-boosting qualities (62.5%), and significance for child health and development (80%). However, there were some misconceptions, with several respondents incorrectly associating VIT D with preventing stroke (10%), diabetes (10%), high blood pressure (9%), and arthritis (40%) (Table 2).

The majority of mothers (74%) demonstrated poor knowledge, scoring 50%. However, 26% of the mothers exhibited good knowledge, scoring above 50% on the section regarding VIT D and its supplementation, with a median score of 11 (*Siddiqee et al., 2021*) (Fig. 1).

**Table 1  Demographic characteristics of mothers studied.**

| Parameters | | Results |
|---|---|---|
| | Age | 33 (8) |
| Age groups | Less than 25 | 4 (2%) |
| | ≥25<30 | 45 (22.5%) |
| | ≥30<35 | 67 (33.5%) |
| | ≥35 | 84 (42%) |
| Marital status | Married | 195 (97.5%) |
| | Divorced | 5 (2.5%) |
| Educational level | Uneducated | 1 (0.5%) |
| | Secondary | 8 (4%) |
| | High School | 44 (22%) |
| | Bachelor | 126 (63%) |
| | Master's and above | 21 (10.5%) |
| Functional status | Employee | 74 (37%) |
| | Unemployed | 126 (63%) |
| Nationality | Saudi | 158 (75.6%) |
| | Non-Saudi | 42 (21%) |
| Number of children | One child | 33 (16.5%) |
| | Two children | 51 (25.5%) |
| | Three kids | 45 (22.5%) |
| | Four children and more | 71 (35.5%) |
| Your youngest child in months | 1–12 months | 81 (40.5%) |
| | 13–24 Months | 47 (23.5%) |
| | 24 months and more | 72 (36%) |
| Type of residency | Owing house | 114 (57%) |
| | Rented house | 86 (43%) |
| Monthly income | Less than 5,000 riyals per month | 38 (19%) |
| | From 5–less than 10,000 riyals per month | 81 (40.5%) |
| | From 10–20 thousand riyals per month | 53 (26.5%) |
| | More than 20 thousand riyals per month | 28 (14%) |
| How to breastfeed the youngest children | Breast feeding | 44 (22%) |
| | Bottle feeding | 105 (52.5%) |
| | Both | 51 (25.5%) |

**Notes.**

Data were presented as N (%) or Median (IQR).

## Mothers' practices towards their children in relation to VIT D and its supplementation

This study shows that 55.5% of mothers did not expose their children to sunlight. Among those who exposed their children to the sun, the majority (45%) reported exposing the child's entire body.

Regarding the mode of sun exposure, 56.5% of the mothers indicated indirect exposure, such as behind glass. Regarding the timing of sun exposure, the most common response was before 10 o'clock in the morning (44%). Among those who did expose their children

**Table 2  Mothers' knowledge about VIT D in Saudi Arabia.**

| | | |
|---|---|---|
| 1. Have you heard of Vitamin D? | | |
| Yes | 198 (99%) | 1 (0) |
| No | 2 (1%) | |
| 2. Is there vitamin D in breast milk? | | |
| Yes | 115 (57.5%) | |
| No | 16 (8%) | 1 (1) |
| I don't know | 68 (34%) | |
| 3. What is the source you learned about vitamin D? (You can choose more than one answer) | | |
| TV, radio, advertisements | 37 (18.5%) | |
| Social media (Snapchat, Instagram, Facebook...) | 87 (43.5%) | |
| Friends–family | 68 (34%) | |
| Doctor/Nurse | 130 (65%) | 2 (2) |
| Magazines | 22 (11%) | |
| Books | 22 (11%) | |
| From studying at school/university | 56 (28%) | |
| Other | 1 (0.5%) | |
| 4. What are the benefits of vitamin D (you can choose more than one answer) | | |
| Bone health* | 176 (88%) | |
| Boosts immunity* | 125 (62.5%) | |
| Helps a healthy pregnancy* | 62 (31%) | |
| Prevents stroke* | 20 (10%) | |
| Prevents diabetes* | 20 (10%) | |
| Prevents high blood pressure* | 18 (9%) | 5 (3) |
| Prevents arthritis* | 80 (40%) | |
| Prevent depression* | 101 (50.5%) | |
| Important for hair health* | 95 (47.5%) | |
| Prevents obesity and weight gain* | 48 (24%) | |
| Important for skin health* | 58 (29%) | |
| Important for child health and development* | 160 (80%) | |
| 5. What are the sources of Vitamin D? (You can choose more than one answer) 3 | | |
| The sun* | 167 (83.5%) | |
| The food* | 42 (21%) | 1 (0) |
| Food Supplements* | 50 (25%) | |
| I do not know | 2 (1%) | |
| 6.Where is Vitamin D found in food? (You can choose more than one answer) | | |
| Milk and dairy products* | 123(61.5%) | |
| Fruits | 48 (24%) | |
| Vegetables | 52 (26%) | |
| Fish* | 101 (50.5%) | 2 (2) |
| Egg Yolk* | 94(47%) | |
| Liver* | 53 (26.5%) | |
| I do not know | 30 (15%) | |
| Total | 11 (7) | |

**Notes.**
*Correct answer Data were presented as N (%) or Median (IQR).

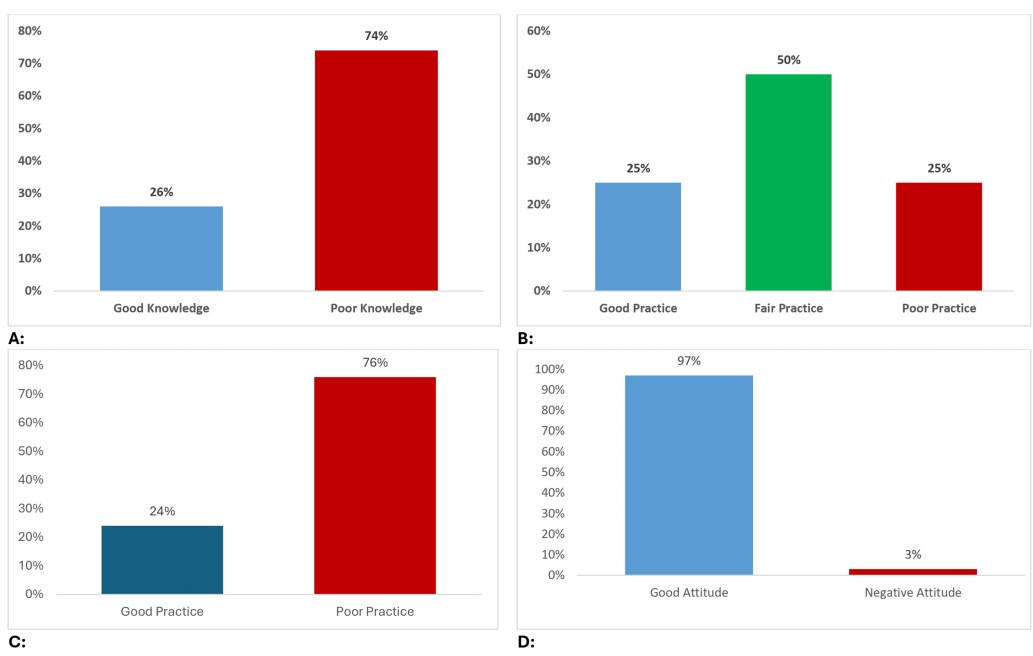

**Figure 1** **Mothers knowledge, attitude and practice towards vitamin D.** (A) Mothers knowledge towards vitamin D and its supplementation. (B) Mothers practice towards vitamin D and its supplementation. (C) Mothers practice towards taking vitamin D and sunlight exposure. (D) Mothers attitude towards vitamin D and its supplementation.

to sunlight, the majority (41.5%) reported exposure for less than an hour per week, while 32% of respondents reported exposure from 1 to 2 h per week. In terms of knowledge about the appropriate dose for sun exposure, 31.5% of mothers reported being aware of the recommended dose, while the remaining 68.5% expressed a lack of knowledge in this regard. Most mothers (61.5%) reported providing their children with fortified VIT D, with 24.5% always committed to giving their children the daily fortified dose while 27.0% did not follow this practice (Table 3). Only 28% of the mothers demonstrated good practices (scores of 75% or higher), while a larger proportion (42%) had fair practices (scores 50–<75%), and 30% had poor practices scores (<50%). The median score was 9 (*Ahmed et al., 2017*) (Fig. 1).

## Mothers' practices towards taking VIT D and sunlight exposure

Mothers' practices towards VIT D revealed inconsistencies, with over half (56%) not taking supplements currently. Reliance on supplementation further decreased during pregnancy (49% reported "sometimes") and breastfeeding (37% reported "sometimes"). Concerns about dietary VIT D were substantial, with 44% of mothers permanently concerned. Daily sun exposure was ensured by only 33.5% of mothers, and among those exposed, most received insufficient amounts (60% with less than an hour weekly). The preferred time for sun exposure was before 10 am (60%) (Table 4). Only 47% of the mothers showed good practices (scores of 50% or higher) while 53% of the mothers showed poor practices (scores < 50%), with a median score of 13 (*Ahmed et al., 2017*) (Fig. 1).

**Table 3  Mothers practices towards their children in relation to VIT D and its supplementation in Saudi Arabia.**

| | | |
|---|---|---|
| 1. I expose my child to the sun daily | | |
| Yes | 89 (44.5%) | |
| No | 111 (55.5%) | 0.5 (1) |
| 2. What part of the child's body exposed to the sun? | | |
| I do not expose my child to the sun | 47 (23.5%) | |
| Face only | 4 (2%) | |
| Hands only | 9 (4.5%) | 2 (2) |
| Feet and legs | 50 (25%) | |
| All the body | 90 (45%) | |
| 3. Do you expose your child to the sun? | | |
| Directly | 87 (43.5%) | |
| Indirectly (Behind the glass, for example) | 113 (56.5%) | 0.5 (1) |
| 4. What time do you expose your child to the sun? | | |
| I do not expose my child to the sun | 39 (19.5%) | |
| Before ten in the morning | 88 (44%) | |
| Between 10 in the morning and 3 in the afternoon | 29 (14.5%) | 2 (1) |
| After three in the afternoon | 33 (16.5%) | |
| Other times | 11 (5.5%) | |
| 5. What is the average number of hours your child is exposed to the sun per week? | | |
| I do not expose my child to the sun | 42 (21%) | |
| Less than an hour | 83 (41.5%) | |
| From one to two hours | 64 (32%) | 1 (1) |
| 6 h and more | 11 (5.5%) | |
| 6. Do you know the appropriate dose for exposing your child to the sun? | | |
| Yes | 63 (31.5%) | |
| No | 137 (68.5%) | 0.5 (1) |
| 7. Do you give your child fortified doses of vitamin D (oral drops)? | | |
| Yes | 123 (61.5%) | |
| No | 77 (38.5%) | 1 (1) |
| 8. Commit to giving my child a daily fortified dose of vitamin D | | |
| Always | 49 (24.5%) | |
| Sometimes | 77 (38.5%) | |
| Scarcely | 22 (11%) | 2 (2) |
| Do not apply | 52 (26%) | |
| 9. What are the reasons for not giving your child the supportive dose of vitamin D? | | |
| Forget | 92(46%) | |
| Child's illness | 6 (3%) | |
| Because I expose it to the sun, there is no need to take vitamin D doses | 39 (19.5%) | |
**Table 3** (*continued*)

| | | |
|---|---|---|
| Because the child gets a dose of vitamin D from formula or Cereal | 28 (14%) | |
| | | 0.5 (1) |
| Mother's preoccupation with other children | 17 (8.5%) | |
| Because the mother is busy with the job | 15 (7.5%) | |
| Because vitamin D doses are not necessary | 7 (3.5%) | |
| Other reasons (The child does not need vitamin D) | 48 (24%) | |
| Total | | 9 (6) |

## Mothers' attitudes towards VIT D and its supplementation

This study revealed a generally favorable attitude among mothers regarding VIT D and sun exposure, both for them and their children. The majority of mothers endorsed VIT D supplementation during pregnancy (80%) and breastfeeding (84.5%), with nearly all of them (97%) favoring blood tests to monitor VIT D levels. Notably, there were high proportions of mothers who supported giving their child VIT D starting at birth (85%) and daily sun exposure (84.5%) (Table 5). Interestingly, 95% of the mothers showed a good attitude (scores of 50% or higher) while only 5% of the mothers showed a negative attitude (<50%) with a median score of 7 (*Bischofova et al., 2018*) (Fig. 1).

## Relationship between demographic factors and knowledge, practice, and attitude

A significant relationship was found between knowledge and the education level ($p = 0.026$), employment status ($p = 0.029$), and nationality ($p = 0.001$) of the mothers. The practice of mothers towards their children was significantly correlated with type of child feeding ($p = 0.016$) while the mothers' VIT D and sun exposure practices were significantly correlated with their employment status ($p = 0.05$). Educational level ($p = 0.034$), employment status ($p = 0.029$), and type of residency ($p = 0.006$) were the most significant factors affecting mothers' attitudes. On the other hand, age, marital status, number of children, and income did not show a significant relationship with knowledge, practice, and attitude of mothers (Table 6).

## DISCUSSION

This study assessed mothers' perceptions of VIT D supplementation for them and their children in Saudi Arabia. The findings revealed that most of the participants had heard about VIT D, and almost two-thirds of them were aware of the sources of VIT D in breast milk. It also showed that exposure to sunlight and milk intake are the main resources of VIT D, and a large proportion of the study sample supported VIT D use during pregnancy and lactation. This was in alignment with the national recommendation for adding VIT D and other minerals to specific food products for high-risk groups such as children, pregnant women, and elderly people (*SFDA, 2022*).

However, greater than two-thirds of mothers reported that, beyond social media, doctors and nurses were information sources regarding VIT D. Our results are consistent with those of a recent study conducted in Egypt (*Soliman et al., 2020*). This similarity indicates that

**Table 4  Mothers' practices towards taking VIT D and sunlight exposure in Saudi Arabia.**

| | | |
|---|---|---|
| 1. I am currently taking vitamin D supplements | | |
| Yes | 88 (44%) | |
| No | 112 (56%) | 0.5 (1) |
| 2. I took vitamin D supplements during my pregnancy | | |
| Always | 30 (15%) | |
| Sometimes | 98 (49%) | |
| Scarcely | 23 (11.5%) | 2 (1) |
| never | 49 (24.5%) | |
| 3. I take vitamin D while breastfeeding | | |
| Always | 36 (18%) | |
| Sometimes | 74 (37%) | |
| Scarcely | 14 (7%) | 2 (2) |
| never | 76 (38%) | |
| 4. I was concerned that the food I prepared during the week contained vitamin D | | |
| Always | 15 (7.5%) | |
| Sometimes | 60 (30%) | |
| Scarcely | 37 (18.5%) | 1 (2) |
| Permanently | 88 (44%) | |
| 5. I exposed to the sun while out of the house | | |
| Yes | 175 (87.5%) | |
| No | 24 (12%) | 1 (0) |
| 6. I make sure that the food I eat during the week contains vitamin D | | |
| Yes | 93 (46.5%) | |
| Sometimes | 68 (34%) | 1 (1) |
| No | 39 (19.5%) | |
| 7. I make sure to be exposed to the sun daily | | |
| Yes | 67 (33.5%) | |
| No | 133 (66.5%) | 0.5 (1) |
| 8. How many hours per week are you exposed to the sun? | | |
| less than an hour | 120 (60%) | |
| From one to two hours | 70 (35%) | 0.5 (1) |
| 6 h and more | 10 (5%) | |
| 9. What parts of the body are exposed to the sun? | | |
| Face | 12 (6%) | |
| Hands | 2 (1%) | |
| Face and hands | 94 (47%) | 2 (1) |
| Face, hands, and legs | 92 (46%) | |
| 10. What is the right time to be exposed to the sun? | | |
| Before ten in the morning | 120 (60%) | |
| From ten to three in the evening | 42 (21%) | 2 (1) |
| After three o'clock in the evening | 38 (19%) | |

*(continued on next page)*

**Table 4** (*continued*)

| 11. Do you use sunscreen when you go out in the sun? | | |
|---|---|---|
| Yes | 83 (41.5%) | |
| Sometimes | 23 (11.5%) | 0.5 (1) |
| No | 94 (47%) | |
| 12. How often do you use sunscreen while out in the sun? | | |
| Always | 47 (23.5%) | |
| Sometimes | 60 (30%) | 2 (2) |
| Scarcely | 43 (21.5%) | |
| Permanently | 50 (25%) | |
| 13. Is there a conflict between applying sunscreen and exposure to the sun? | | |
| Yes | 22 (11%) | |
| No | 104 (52%) | 0.5 (0) |
| I do not know | 74 (37%) | |
| Total | | 13 (6) |

**Table 5** Mothers' attitudes towards VIT D and its supplementation in Saudi Arabia.

| 1. Do you support taking vitamin D during pregnancy? | | |
|---|---|---|
| Yes | 160 (80%) | |
| No | 40 (20%) | 1 (0) |
| 2. Do you support taking vitamin D during breastfeeding? | | |
| Yes | 169 (84.5%) | |
| No | 31 (15.5%) | 1 (0) |
| 3. Do you support a blood test to detect the level of vitamin D in the body? | | |
| Yes | 194 (97%) | |
| No | 6 (3%) | 1 (0) |
| 4. If the answer to the previous question is yes, do you suggest that it be? | | |
| Periodically | 80 (40%) | |
| From time to time | 94 (47%) | 2 (1) |
| At the request of the doctor | 25 (12.5%) | |
| 5. Do you support giving your child vitamin D since birth? | | |
| Yes | 170 (85%) | |
| No | 30 (15%) | 1 (0) |
| 6. Do you support exposing your child to the sun daily? | | |
| Yes | 169 (84.5%) | |
| No | 31 (15.5%) | 1 (0) |
| 7. Do you support the use of sunscreen while exposing your child to the sun? | | |
| Yes | 87 (43.5%) | |
| No | 113 (56.5%) | 0.5 (1) |
| Total | | 7 (2) |

**Table 6 Relationship between demographic factors and knowledge, practice, and attitude of Vit D.**

| Parameters | Mothers knowledge | Mothers practice related to children | Mothers practice towards taking vitamin D and exposure to sunlight | Mothers attitudes |
|---|---|---|---|---|
| | | | *P*-value | |
| Age | 0.103 | 0.949 | 0.904 | 0.287 |
| Marital | 0.514 | 0.742 | 0.432 | 0.119 |
| Education | 0.026 | 0.229 | 0.614 | 0.034 |
| Functional | 0.029 | 0.910 | 0.050 | 0.029 |
| Nationality | 0.001 | 0.634 | 0.769 | 0.246 |
| Number of children | 0.427 | 0.318 | 0.966 | 0.221 |
| Youngest child | 0.135 | 0.806 | 0.668 | 0.826 |
| Living | 0.988 | 0.627 | 0.447 | 0.006 |
| Income | 0.722 | 0.071 | 0.994 | 0.972 |
| Feeding | 0.781 | 0.016 | 0.184 | 0.873 |

healthcare providers are the main source of information to increase and promote a patient's adequate understanding of the important role of VIT D.

In regards of VIT D benefits, the majority of mothers reported that bone health is the most important benefit of VIT D, and more than two-thirds of them confirmed the important role VIT D plays in the development of child health. Most mothers agreed that the sun is the major source of VIT D, followed by food supplements. Almost two-thirds of the mothers confirmed that milk and dairy products are the most important food sources of VIT D. Moreover, the results of our study revealed that more than two-thirds of the mothers had poor knowledge and more than one-third of them had good knowledge about VIT D and its supplementation. These results also coincide with those of a study carried out in Iraq (*Rasheed, Taha & Rasheed, 2017*), which could be due to the similarity in the samples studied.

Regarding mothers' practices towards their children in relation to VIT D and its supplementation, the results of our study showed that greater than half of the participants did not expose their child to the sun daily. Less than half of the participants reported exposing their child's entire body to the sun, while less than one quarter of them did not expose their child to the sun at all. More than half of mothers reported that they exposed their child indirectly to the sun before 10 o'clock in the morning. As for the average exposure of children to weekly sun, less than half of mothers reported that they exposed their child to sun for less than one hour weekly, and almost one-third of them exposed their child to sun between 1 to 2 h daily. At the same time, more than two-thirds of mothers in this study did not know the appropriate dosage when exposing children to sunlight. These findings are also in agreement with the findings of a study carried out in Malaysia (*Hussein et al., 2018*).

Moreover, our results showed that there was interest and awareness regarding the important role of VIT D for children, as more than two-thirds of studied mothers give a daily fortified dose of VIT D, more than one-third reported that they are "sometimes"
committed to giving oral VIT D drops, and almost one-quarter of mothers reported that they are "always" committed to giving oral drops. Regarding the reasons for not giving their children the recommended dose of VIT D, almost half of the mothers reported that they forgot, and less than one-third of them reported that they expose their child to the sun daily so there is no need to give a supplemental VIT D oral dose. These results are supported by a previous study (*Hussein et al., 2018*) that reported that nearly all of the study sample exposed their children to the sun. However, almost half of the children were fully covered during sun exposure, with an average time of twenty minutes of exposure per day, and most of the participants exposed their children to the sun early in the morning before 10 am. Moreover, our present study indicates that a minority of mothers had fair practices toward VIT D and its supplementation, while less than one-quarter of them had good practices, and more than one-third of mothers had poor practices. These results may be due to more than two-thirds of our sample poor knowledge of VIT D and its supplementation.

Regarding mothers' attitude toward VIT D and its supplementation, the present study indicated that more than three-quarters of the mothers reported that they used VIT D during pregnancy. Additionally, the majority of mothers reported taking VIT D during breastfeeding, which indicates a high level of interest in and awareness of VIT D during pregnancy and breastfeeding. Almost all the participants in our study support the use of blood tests to detect the level of VIT D in the body. Furthermore, our study results found that more than three-quarters of the mothers reported that they supported giving their child VIT D starting at birth. However, our study findings indicated that two-thirds of mothers agreed to expose their child to the sun daily, and almost two-thirds of them supported the use of sunscreen while exposing their child to the sun. The current study revealed that most mothers had a positive attitude towards VIT D and its supplementation. This may be due to the majority of the participants previously hearing about VIT D, which may have contributed to their positive attitudes toward VIT D. Our study findings aligned with those of *Dağhan et al. (2019)*, which identified prevalent practices of mothers who used VIT D during their pregnancies administering oral VIT D supplements to their infants and facilitating sun exposure for their children. Moreover, our findings were also congruent with a recent study conducted in Saudi Arabia by *Babelghaith et al. (2017)*, who revealed that most of the studied sample were willing to do a VIT D blood test.

Regarding the practices of mothers toward taking VIT D and sunlight exposure, the current results found that greater than half of the participants did not take VIT D supplements, while almost have of them reported that they "sometimes" used VIT D supplements during their last pregnancy.

As for the breastfeeding period, more than one-third of mothers reported taking VIT D "sometimes" or "never". Moreover, more than one-third of mothers in this study reported that they were "never" concerned about food rich in VIT D. Nearly all of the participants reported that sun exposure was common during outdoor activities. More than two-thirds of mothers were not exposed to sunlight daily, and were exposed less than 1 h per week. Half of the mothers reported exposing their hands and face, and more than two-thirds

of them were exposed to the sun before 10 o'clock in the morning. Only 18.4% of the participants reported that between 10 am and 3 pm was the best time for sun exposure.

Finally, our results revealed that more than half of mothers "sometimes" used sunscreen, and less than half of them did not believe that there was a conflict between the use of sunscreen and exposure to sunlight. Moreover, this study's findings suggest that a substantial proportion of participants demonstrated moderately appropriate behaviors regarding VIT D, while one-third of them had poor practices, and fewer than one-quarter of mothers had good practices towards VIT D and its supplements. The current study's findings aligned with those of research conducted in India (2015), which reported limited VIT D supplementation among antenatal mothers, while also revealing that a significant portion received daily sun exposure exceeding 1 h. The majority of mothers did not know the best time for sun exposure, and they were not concerned about the VIT D content of food, and reported other poor practices (*Kavitha, 2015*). These similar results might be attributed to the mothers having limited knowledge of VIT D sources and its importance.

Regarding the factors affecting mothers' knowledge, attitudes, and practices toward VIT D supplementation and importance, our study results indicate that there was a statistically significant correlation between mothers' attitudes towards VIT D and both their educational level and functional status. Mothers who were employed and had higher level education had more positive attitudes than others in the sample. This study identified a statistically significant correlation between mothers' monthly income and their knowledge of VIT D and its supplements.

Mothers with higher incomes demonstrated greater knowledge in this area. Additionally, the analysis revealed a statistically significant link between mothers' age within the sample and their practices related to VIT D and supplementation. This finding aligns with prior research by *Aljefree, Lee & Ahmed (2017)* and *Soliman et al. (2020)*, highlighting the influence of sociodemographic factors, such as age and income, on mothers' behaviors and practices regarding VIT D supplementation.

## CONCLUSION

The present investigation found that a large proportion of the studied population had clear knowledge of VIT D and its important role for infants. The results of the study also showed that mothers are aware that sunlight exposure and milk intake are the main sources of VIT D, and a large proportion of the study sample supported VIT D use during pregnancy and lactation. Sociodemographic factors, including monthly income, age, educational level, and employment status, were found to be significantly associated with mothers' practices and attitudes towards VIT D.

## RECOMMENDATIONS

The findings highlight the significance of assessing mothers' knowledge, attitudes, and practices when administrating VIT D to their children. For mothers, ongoing training programs are recommended to enhance levels of knowledge and practices about VIT D, as well as improving awareness and practice of incorporating daily sources of VIT D

and supplements. The researchers also recommend the need to conduct further extensive studies on other societies, groups, and ages to find out how important VIT D is to those communities. It is also highly recommended to conduct further research to identify the correlation between mothers' knowledge of VIT D and children's health status, and its consequences on child health and immune systems.

## STRENGTHS AND LIMITATIONS

The results of this study add to current research on VIT D supplementation use for mothers and their children, especially in Saudi Arabia. Furthermore, the participants were drawn from different cities in Saudi Arabia, which provide a general picture on mothers' perceptions of VIT D supplementation for them and their children. The results could be resourceful for health organizations to develop guidelines on VIT D supplementation and educational programs to improve mothers' knowledge and practice of VIT D. On the other hand, the study has some limitations including using a convenience sample on a selected number of mothers of children under five years which could result in lack of generalizability of the findings.

### Funding

This work was supported by Princess Nourah bint Abdulrahman University Researchers Supporting Project number (PNURSP2025R441), Princess Nourah bint Abdulrahman University, Riyadh, Saudi Arabia. The funders had no role in study design, data collection and analysis, decision to publish, or preparation of the manuscript.

### Grant Disclosures

The following grant information was disclosed by the authors:
Princess Nourah bint Abdulrahman University Researchers Supporting Project: PNURSP2025R441.

### Competing Interests

The authors declare there are no competing interests

### Author Contributions

- Hanan F. Alharbi conceived and designed the experiments, performed the experiments, analyzed the data, prepared figures and/or tables, authored or reviewed drafts of the article, and approved the final draft.
- Khulud Ahmad Rezq conceived and designed the experiments, performed the experiments, analyzed the data, prepared figures and/or tables, authored or reviewed drafts of the article, and approved the final draft.

## Ethics

The following information was supplied relating to ethical approvals (*i.e.*, approving body and any reference numbers):

Institutional Review Board (IRB) at Princess Nourah bint Abdulrahman University in Riyadh, Saudi Arabia (HAP-01-R-059).

## Data Availability

The raw data is available in the Supplemental File.

## Supplemental Information

Supplemental information for this article can be found online at http://dx.doi.org/10.7717/peerj.19781#supplemental-information.

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
