# Peer review of "Vitamin D supplementation in Saudi Arabia: mothers’ knowledge, practices, and attitudes"

_PeerJ, doi:10.7717/peerj.19781_

## Round 0.1 · original submission · Major Revisions

Dear Dr. Alharbi,

Thank you for submitting your manuscript entitled "Vitamin D supplementation in Saudi Arabia: mothers' knowledge, practices, and attitudes: A Cross-Sectional Study" to our journal. After careful consideration of the reviewers' comments and my own evaluation, I have decided that your manuscript requires major revisions before it can be considered for publication.

The study addresses an important topic with potential implications for public health in Saudi Arabia. However, there are several critical issues that need to be addressed:

Methodological clarity: Please provide more detailed information about participant recruitment, selection criteria, and statistical analyses.
Data presentation: Ensure consistency between the methods, results, and discussion sections. Improve the presentation of figures and tables as suggested by the reviewers.

Context and background: Strengthen the introduction with more comprehensive information on vitamin D deficiency prevalence and national recommendations in Saudi Arabia.

Language and structure: A thorough English language review is necessary to improve readability and clarity throughout the manuscript.

Conclusions and implications: Strengthen the discussion of practical implications and address the limitations of the study, particularly regarding the representativeness of the sample and the lack of data on children's health status.

We believe that addressing these issues will significantly enhance the quality and impact of your manuscript.

We look forward to receiving your revised submission.

Reviewer 1 ·

Basic reporting

Thank you for the opportunity to read this manuscript, which addresses an important topic and provides valuable insights into maternal knowledge, attitudes, and practices regarding vitamin D supplementation. However, there are several important aspects that need to be addressed. I hope that my comments will be helpful in developing the manuscript.

1. The title clearly introduces the reader to the topic and explains the study design. The abstract is informative and provides a summary of what was found. I suggest adding information in the abstract about when the data were collected and a brief description that an online questionnaire was used to assess knowledge, practices and attitudes towards vitamin D supplementation.
2. Lines 62-66: I suggest you emphasise that vitamin D deficiency is common in more age groups than in children 5-10 years and support this with more references. I also suggest that you include information on the prevalence of vitamin D deficiency/insufficiency in the KSA.
3. Lines 65-66: Add reference to support for this information.
4. Line 67: Reference Alshahrani (2014) is not found in the reference list.
5. Lines 71-73: The information in the sentence can't be found in the article to which the sentence refers (Tønnesen, R., Peter, H., Thorbjorn, J., & Peter, S. (2017)).
6. I suggest that you include information on any national recommendations for vitamin D supplementation or vitamin D fortification programme in the KSA, in order to provide the reader with a more comprehensive context.
7. Figure 1: I suggest that you split the different figures in Figure 1 into Figures 1a, 1b, 1c and 1d to improve the flow of the text. I think that Figure 1 shows the proportion of mothers who scored in a particular way? I suggest that the figure legend should include more information to help the reader understand the figure. It is also not clear what is meant by the numbers on the y-axis in the bar chart at the bottom and on the left, and whether the numbers on the x-axis should be converted to percentages as in the other bar charts.
8. Table 1: The category “Your youngest child in months” is shown in years in the next column.
9. Table 6: Table 6 shows the results of the ANOVA test, correct? The heading could be misleading, because you are testing whether there are statistically significant differences between the means of the groups rather than analysing the relationship between demographic factors and knowledge, practice and attitudes.
10. At several parts of the manuscript some words and/or letters are missing or doubled, which disrupts the flow of the reading. Some examples are at line 29: themselves (instead of them); line 37: with (2 times); line 42: proportion (instead of portion); line 44: the sentence missing a verb (was demonstrated?); lines 98-103: two sentences with identical information; table 1 heading: demographic (instead of demographical), add “of” the mothers studied.

Experimental design

The research question is relevant and meaningful, and the manuscript provides information on the knowledge gap and how this study helps to fill it. However, several parts need further explanation:
11. Information about the recruitment of the mothers is lacking. Was the recruitment also performed via social media platforms or was it limited to the data collection?
12. Lines 94-95: You stated that those whose youngest child was older than 5 years were excluded. However, in the results section (line 171) you show that most mothers have a youngest child aged between one and 12 years, which statement is correct?
13. Lines 143-147: What were the null and alternative hypotheses on which the power calculation was based? I assume that there is more information about the power calculation in the reference Faul et al. (2007), but it is not included in the reference list.
14. Lines 150-151: You stated that the data were presented as mean and standard deviation, but results section (line 167) shows medians.
15. Thank you for providing the raw data, it was clearly structured which made it easy to follow. However, more information on the statistical analysis is needed. Are you testing for differences in mean scores for the knowledge, practice and attitude sections between different age groups and different education levels of the mothers?
16. Line 167: How did the study sample arrive at? Were there more mothers than the 200 included who completed the questionnaire but were not eligible for inclusion?

Validity of the findings

17. The information on how the data were analysed needs to be improved so that the information in the methods is consistent with the information described in the results and discussion. The results of the correlations between vitamin D and maternal demographic factors are presented at lines 37-40, 222-225, and 316-319. However, it is unclear whether the manuscript includes an analysis of these correlations (not stated in the data analysis section). Lines 221, 332-334: You describe that you found a significant relationship and association between mothers’ knowledge, and mothers’ level of education, employment status, and nationality. I suggest that you rewrite these sentences as it is not clear that this is the result of the ANOVA test. The sentences could be misinterpreted as you calculated correlation coefficients or analysed the determinants of mothers’ vitamin D knowledge by regression analysis (which could be another way of analysing the data).
18. Lines 231-234: I suggest that you summarise the key findings early in the discussion. Make a clear link between the findings and the aim of the study. In addition, the discussion can be improved by placing your findings in a broader context, for example by highlighting why knowledge should be improved and comparing your findings with national recommendations on vitamin D supplementation and sun exposure.
19. Lines 327-328: You stated that “The results of the study also showed 328 that exposure to sunlight and milk intake are the main resources of VIT D”, but this is not consistent with the aim of the study. Further, it is not possible to draw a conclusion regarding the main sources of vitamin D based on the data presented in the manuscript. My guess is that this sentence is about where mothers have answered where vitamin D can be found in food, correct?
20. Lines 329-332: This information does not align with the aim of the study.
21. Lines 336-337: The sentence may be open to misinterpretation, as it appears to imply that maternal vitamin D levels (25-hydroxyvitamin D) should be measured.
22. Line 343-350: I suggest that you extend the discussion of whether this is a population-representative study population, given the relatively small size of the study sample. It would be helpful to ascertain whether the study population aligns with the general population in terms of age, education level, and other relevant characteristics.

Reviewer 2 ·

Basic reporting

The use of English is mostly correct throughout the manuscript, although editing is certainly required, i.e. line 44 “However, a positive attitude towards VIT D supplementation among the majority of mothers” lacks the main verb and line 60 “Moreover, VIT D deficiency has been reported to the development of cardiomyopathy” is not a grammatically correct sentence. Other examples include but are not limited to lines 80, 97, 181, 238, 268, 297. I suggest you have a colleague who is proficient in English and familiar with the subject matter review your manuscript, or contact a professional editing service.
The manuscript is structured in an appropriate and rational way. The background is well-detailed and references cited in this section are appropriate and relevant to the matter.
Tables and figures are clear and understandable. I would only recommend improving the layout for table 6, as there is much information which might be hard to read in the current layout (for example, by including only the p-value and eliminating the “p”).
I appreciate the fact that the authors shared their raw dataset and reported the survey questions in the tables.

Experimental design

The research question is clearly stated and the authors have described how this research fits in the current knowledge on the matter. The idea for the research question is creative and smart.
Regarding the study design, please explain the reason for the selection of mothers with children under the age of 5 years, as it is not explained in the manuscript: did you base your decision on other works in the literature or is 5 years of age an important cutoff for some reason?
The study design is appropriate and sensible; I would like the authors to add some technical information regarding the google form: was it anonymous or not? And, if it was anonymous, was there any way to prevent the same user from answering more than once? Also, details about possible selection bias, since the form was only available through the internet, should be discussed in the materials and methods as well as the strengths and limitations sections.
Statistical analysis was carried out in an appropriate way.

Validity of the findings

Results are described in a clear way and the discussion is quite detailed, with up-to-date and relevant references.
Conclusions are well stated and linked to original research question. However, I am left wondering what the practical repercussions of said results are for everyday clinical practice, except for the need of training programs which were already seen as important a priori in the background: this should be addressed before acceptance. Also, in the conclusion, the sentence “the need to guide this group about the important role of taking VIT D and taking nutritional supplements rich in it because of its great importance that reflects on the health of infants” is not sufficiently backed up by data, as actually in this research no data regarding the real health of the children involved in the study were gathered (i.e., vitamin D levels, calcium levels, and so on).
It would indeed be interesting to investigate whether there is a correlation between mothers’ knowledge of vitamin D and children’s health status, i.e. if the mothers who have poor practices regarding vitamin D have children with poorer bone health or immune system deficiencies: this should be mentioned in the discussion and conclusion.

Additional comments

In general, this is an interesting and well-researched research article with some flaws which can be improved.

---

## Round 0.2 · Minor Revisions

We regret that we have to ask authors to address general language issues prior to acceptance. We recommend that authors consult with a colleague who is familiar with the subject matter and who is fluent in English, or use a professional editing service.

**Language Note:** The Academic Editor has identified that the English language must be improved. PeerJ can provide language editing services - please contact us at [email protected] for pricing (be sure to provide your manuscript number and title). Alternatively, you should make your own arrangements to improve the language quality and provide details in your response letter. – PeerJ Staff

---

## Round 0.3 · accepted · Accept

Thank you to the authors for addressing all the comments. I am now happy with the current version. The manuscript is ready for publication.